# Paternal methotrexate exposure affects sperm small RNA content and causes craniofacial defects in the offspring

Nagif Alata Jimenez[1,2], Mauricio Castellano[3,4], Emilio M. Santillan [5], Konstantinos Boulias[6,7], Agustín Boan[1,2], Luisa F. Arias Padilla [1,2], Juan I. Fernandino[1,2], Eric L. Greer [6,7], Juan P. Tosar [3,4], Luisa Cochella [5] & Pablo H. Strobl-Mazzulla [1,2] ✉

Folate is an essential vitamin for vertebrate embryo development. Methotrexate (MTX) is a folate antagonist that is widely prescribed for autoimmune diseases, blood and solid organ malignancies, and dermatologic diseases. Although it is highly contraindicated for pregnant women, because it is associated with an increased risk of multiple birth defects, the effect of paternal MTX exposure on their offspring has been largely unexplored. Here, we found MTX treatment of adult medaka male fish (*Oryzias latipes*) causes cranial cartilage defects in their offspring. Small non-coding RNA (sncRNAs) sequencing in the sperm of MTX treated males identify differential expression of a subset of tRNAs, with higher abundance for specific 5' tRNA halves. Sperm RNA methylation analysis on MTX treated males shows that m5C is the most abundant and differential modification found in RNAs ranging in size from 50 to 90 nucleotides, predominantly tRNAs, and that it correlates with greater testicular *Dnmt2* methyltransferase expression. Injection of sperm small RNA fractions from MTX-treated males into normal fertilized eggs generated cranial cartilage defects in the offspring. Overall, our data suggest that paternal MTX exposure alters sperm sncRNAs expression and modifications that may contribute to developmental defects in their offspring.

Folate is a water-soluble vitamin obtained from the diet that is essential for vertebrates. It is incorporated as an essential cofactor for the synthesis of nucleotides and the generation of S-adenosylmethionine (SAM) which serves as a universal donor of methyl groups for DNA, RNA and proteins implicated in gene regulation during early development[1–4]. Maternal folate deficiency leads to severe neural tube defects and craniofacial anomalies of descendants[5–7]. Importantly, the prevalence of these defects is highly reduced by folic acid supplementation prior and during pregnancy[8,9]. Despite global efforts to supplement the maternal diets with folate, there is still a worldwide prevalence of these congenital defects[10–12]. Methotrexate (MTX) is a recognized teratogenic folic acid antagonist that has been linked to an elevated incidence of congenital anomalies in children born from exposed women. Intrauterine MTX exposure has been linked to craniofacial and limb defects, as well as developmental delays[13,14]. In addition to oral clefts, folic acid antagonists may raise the risk of

[1]Laboratory of Developmental Biology, Instituto de Investigaciones Biotecnológicas- Instituto Tecnológico de Chascomús (CONICET-UNSAM), Chascomús, Argentina. [2]Escuela de Bio y Nanotecnologías (UNSAM), Chascomús, Argentina. [3]Functional Genomics Unit, Instituto Pasteur de Montevideo, Montevideo, Uruguay. [4]School of Science, Universidad de la República, Montevideo, Uruguay. [5]Department of Molecular Biology and Genetics, Johns Hopkins University School of Medicine, Baltimore, MA, USA. [6]Department of Pediatrics, HMS Initiative for RNA Medicine, Harvard Medical School, Boston, MA, USA. [7]Division of Newborn Medicine, Boston Children's Hospital, Boston, MA, USA. ✉e-mail: strobl@intech.gov.ar

cardiovascular, neural tube, and urinary tract abnormalities[15]. As a result, current recommendations urge that mothers stop using MTX at least three months before conception[16]. Prior research has also identified a variety of issues concerning MTX use and a probable genotoxic effect on sperm, which might result in chronic disease or congenital anomalies[17]. However, medical care recommendations for males taking MTX while trying to conceive are less clear.

For decades, the sperm genome has been considered transcriptionally quiescent and solely contributing to the restoration of the ploidy of the zygote. However, more recently, a set of functional RNAs have been characterized in mature spermatozoa that are delivered to the oocyte upon fertilization, contributing to early embryo development and thus, influencing the phenotypic outcome of the offspring[18–24]. Intriguingly, paternal folate concentrations can affect the sperm epigenome[25,26]. Whereas the direct impact of these changes is expected to be minimal given the protamine exchange and resetting of DNA methylation during spermatogenesis and early development[27–29], we wondered whether paternal folate levels may also affect the RNA composition of mature sperm.

Small non-coding RNAs (sncRNAs) are a particularly attractive potential carrier of non-genetic information in the spermatozoa. In particular, tRNA-derived small RNAs (tsRNAs) and microRNAs (miRNAs) are the most abundant in mature spermatozoa[30,31]; and have been identified as molecular carriers of paternal experiences, including high fat diet[22,24,32], low protein diet[33], stress[21,34], and odoriferous sensitivity to chemicals[23]. Small RNA biogenesis, stability and functionality are highly dependent on their post-transcriptional modification status, primarily methylation[35–37]. Furthermore, transmission of paternally acquired metabolic disorders is dependent on the presence of post-transcriptional modifications in sperm sncRNAs[19,24].

Here, we explored the intriguing possibility that paternal folate deficiency impacts the offspring's development, and that it may do so through changes in sncRNA abundance and methylation levels. We injected medaka male fish with the folate inhibitor methotrexate (MTX) and characterized their offspring's developmental defects. Next, we analyzed and compared the abundance and modifications of sncRNAs present in the sperm of MTX-treated males to test the idea that they work as mediators of congenital developmental defects.

## Results

### Paternal folate deficiency induced cranial cartilage malformations in their offspring

To investigate the impact of paternal folate deficiency on the development of their progeny, we administered medaka male fish with an intraperitoneal injection of methotrexate (MTX), a well-known folate inhibitor[38–40], at 10 mg of MTX per Kg of body weight (10MTX) and 50 mg/Kg MTX (50MTX)(Fig. 1a). After 7 days, we fertilized wild type oocytes with sperm extracted from treated and untreated males. None of our treatments had a significant impact on sperm fecundity, hatching time, or overall embryo hatching (Fig. S1).

Several studies have shown that folate is an important vitamin for neural and neural crest development in several vertebrate species including humans[5,41,42]. Moreover, maternal folate deficiency during pregnancy leads to abnormal development of neural crest derivatives such as cranial cartilages[43–46]. Taking this into account, we first evaluated the effect of paternal folate deficiency on the development of the offspring's cranial cartilages by performing alcian blue staining at 3 days post hatching-stage (3dph). We measured the length of three dorsal cartilages (anterior orbital, epiphyseal bar and posterior orbital), four ventral cartilages (Meckel, ceratohyal, basibranchial and palatoquadrate), and the Meckel's area and ceratohyal angle (Fig. 1b–j). From the dorsal cartilages, we found a significant reduction in the length of the anterior orbital (also known as taenia marginalis anterior) in the 50MTX group (115.02 µm ± 9.03, one-way ANOVA followed by multiple comparison Tukey's test $p = 0.0164$) when compared to the

10MTX (130.02 µm ± 10.22) and control (125.98 µm ± 15.77). On the ventral side, the basibranchial and Meckel's cartilages were not affected. However, the ceratohyal was reduced to almost half the length, at both 10MTX (192.99 µm ± 7.55, $p < 0.0001$) and 50MTX (185.42 µm ± 8.71, $p < 0.0001$) compared with control (363.64 µm ± 18.96). Interestingly, when we looked at the morphology of those cranial cartilages, we found that two of them, the anterior orbital and basihyal, presented an abnormal shape (Fig. 2). In particular, the anterior orbital has an abnormal serpentine shape, compared with the normal curved shape (Fig. 2a, b). This phenotype was significantly prevalent ($p = 0.0059$) at the 50MTX group (Fig. 2c). However, one of the most drastically affected cartilages was the basihyal, whose phenotypes presented a curved trowel shape (mild) or hook shape (strong) (Fig. 2d). Quantitation of those phenotypes evidences a significant increase in the severity of them at both 10MTX ($p = 0.0329$) and 50MTX ($p = 0.0006$) compared with Control group (Fig. 2e). Overall, these findings support the notion that paternal MTX exposure affects the development of the offspring's cranial cartilage, indicating that sperm may convey some information involved in the observed phenotypic inheritance.

### SncRNAs abundance is altered in the sperm of MTX-treated males

Epigenetic information, including histone modifications and DNA methylation, particularly from the paternal lineage, is largely reprogramed during germline and early embryo development. However, increasing evidence indicates that sncRNAs are a carrier of epigenetic information across generations and may act as mediators of paternally inherited traits[18–23,47]. To assess if paternal folate deficiency affects the small RNA content, we sequenced size selected (~18–30 nt long) RNAs from sperm of 10MTX and control males. Based on the analysis of three biological replicated for each group, we were able to identify different populations of sncRNAs including: ribosomal RNAs (rRNAs), small nuclear RNAs (snRNAs), small nucleolar RNAs (snoRNAs), micro RNAs (miRNAs), transfer RNA fragments (tRNAs), and other miscellaneous RNAs (miscRNA) (Fig. 3a). The comparative analysis showed that rRNAs and tRNA fragments were the most abundant on both control and 10MTX treatment. Some of the most abundant rRNAs mapped to 28S, 18S, and 5.8S rRNA. Reads mapping 5S rRNA and both 16S and 12S mitochondrial rRNA (mtrRNA) were also detected, albeit in a much lower proportion (Fig. 3b). Despite being the most abundant sncRNAs, we found no statistically significant difference in either cytoplasmic or mitochondrial rRNA expression between control and MTX-treated fish (Fig. 3c, d). On the other side, some of the most abundant tRNA fragments, aspartic acid (having the anticodon $Asp^{GUC}$), glutamic acid ($Glu^{CUC}$ and $Glu^{UUC}$), lysine ($Lys^{CUU}$) and glycine ($Gly^{GCC}$) (Fig. 3b), became further significantly enriched upon MTX treatment (Fig. 3c, d). Together, these results demonstrate that paternal MTX exposure affects the relative abundance of specific sncRNAs in the sperm, with tRNA fragments being the most affected population.

### 5′ halves of particular tRNAs are preferentially affected by methotrexate treatment

tRNAs can be cleaved into 5′ and 3′ halves, known as tsRNAs, in response to stress or other external factors[19,24,35,48]. Of particular interest in the sperm RNA content is the large abundance of those tsRNA fragments and their potential regulatory roles in early embryo development[19,20,24,30,31]. Interestingly, we observed that 5′ tsRNA fragments, product of the cleavage of three of the most abundant tRNAs (5′ tsRNA of $Asp^{GUC}$, $Lys^{CUU}$, and $Glu^{CUC}$), were significantly increased in MTX sperm, without a concomitant increase in their respective 3′ tsRNA fragments (Fig. 4a, b, e). Quantification of the proportion of 5′ halves relative to their corresponding 3′ halves showed a significant increase in the percentage of 5′ halves for tRNA $Asp^{GUC}$ and $Gly^{GCC}$ in the MTX-treated samples (Fig. 4c, e). It is interesting to note that for some tRNAs (i.e., tRNA $Glu^{CUC}$, $Glu^{UUC}$, and $Gly^{UCC}$) we mostly retrieve reads

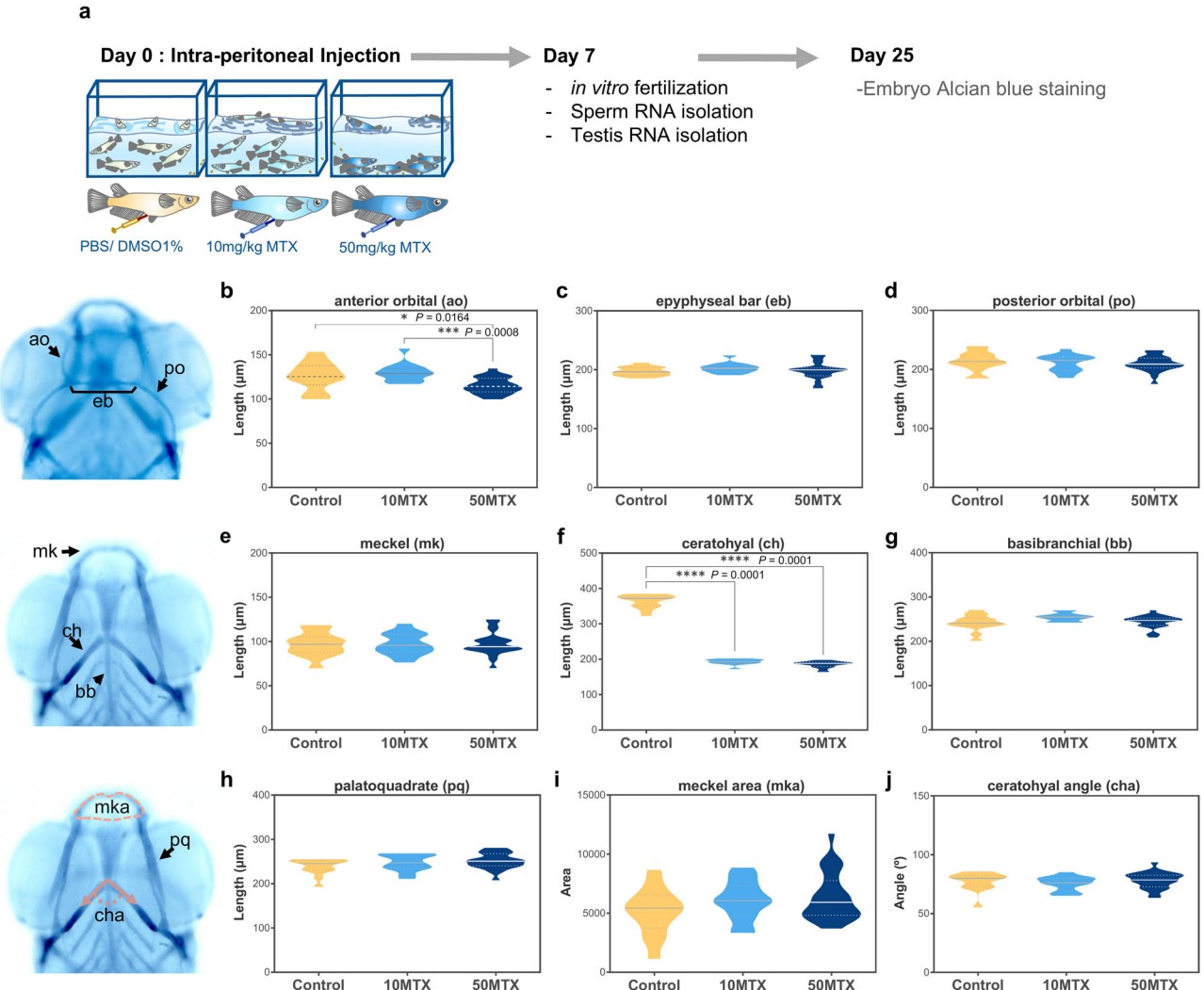

**Fig. 1 | Paternal MTX injection affects offspring's cranial cartilages lengths. a** scheme of experimental design. **b–j** Violin plots represent the measurement of different cranial cartilages lengths, angles and areas on control (Ctrl), 10MTX and 50MTX. Statistical analyses were performed using ANOVA one-way followed by multiple comparison Tukey's test. Numbers of analyzed embryos: Control ($n = 14$), 10MTX ($n = 14$), 50MTX ($n = 26$). *$P = 0.0164$, ***$P = 0.0008$, ****$P = 0.0001$.

for their 5′ halves, but their corresponding 3′ halves are almost undetected for both control and MTX. On the other hand, the 5′ halves of many (most) other tRNAs did not show differences compared with their 3′ halves (tRNA Pro$^{UGG}$, Arg$^{UCU}$), or a major proportion of their 3′ halves (tRNA Ser$^{GCU}$) (Fig. S2). These results suggest changes in processing or stability of specific tRNA fragments as a consequence of the MTX treatment.

Production of 5′ tsRNAs fragments in the 15–22 nt range occurs in multiple tissues and cell lines[49], whereas longer 5′ tsRNAs (31–40 nt long) are preferentially generated in response to different stresses[50,51]. We thus compared the length distribution of tRNAs-derived fragments in both conditions and observed a shift towards longer fragments in 10MTX relative to control samples (median length of 24 nt for control and 29 nt for MTX) (Fig. 4d). A similar situation is found for the tRNAs Glu$^{CUC}$, Asp$^{GUC}$ and Lys$^{CUU}$ where their 5′tsRNAs are significantly increased and their coverage lengths are greater in MTX-treated samples than in controls (Fig. 4f). There is a chance that bias size selection occurred when separating the small RNAs from the gel, resulting in these differences. However, we found that the read coverage and size distribution for the most abundant rRNA-derived fragments are slightly larger after MTX treatment (Fig. S3a), but this is not as dramatic as observed for tsRNAs (median length of 22 nt to 24 nt for rRNAs, vs. 24 nt to 29 nt for tsRNAs; Fig. S3b). These findings suggest

that paternal MTX exposure alters the abundance and cleavage site of specific 5′tsRNAs in the sperm.

## m5C modifications are increased by methotrexate

Post-transcriptional modifications of tRNAs, including methylation, are important for their specific cleavage, stability, and functionality, as well as for the transmission of paternal experiences to the offspring[35–37,52]. We thus evaluated the methylation status of two populations of RNAs isolated from polyacrylamide: a 20–50 nt population (mostly enriched for miRNAs and tsRNAs), and a 50–90 nt population (mostly enriched for mature tRNAs). Within the 20–50 nt RNA population we did not observe significant differences in the abundance of any of the analyzed methylation events between MTX and control groups (Fig. 5a). Conversely, within the 50–90 nt population, we found that MTX treatment led to a significant increase in the relative abundance of several modifications (Fig. 5b). From the two most abundant modifications analyzed (m1A and m5C) only m5C was significantly increased ($p = 0.0155$) in MTX treated samples. From the other less abundant modifications only m2G, m7G and m2′2G presented a significant increase in MTX samples ($p$ value = 0.0027; 0.0063; and 0.0090, respectively). Interestingly, the most abundant modification observed to be differentially detected in MTX samples has been described to be located at the 3′ ends of tRNAs[53,54] (Fig. 5c).

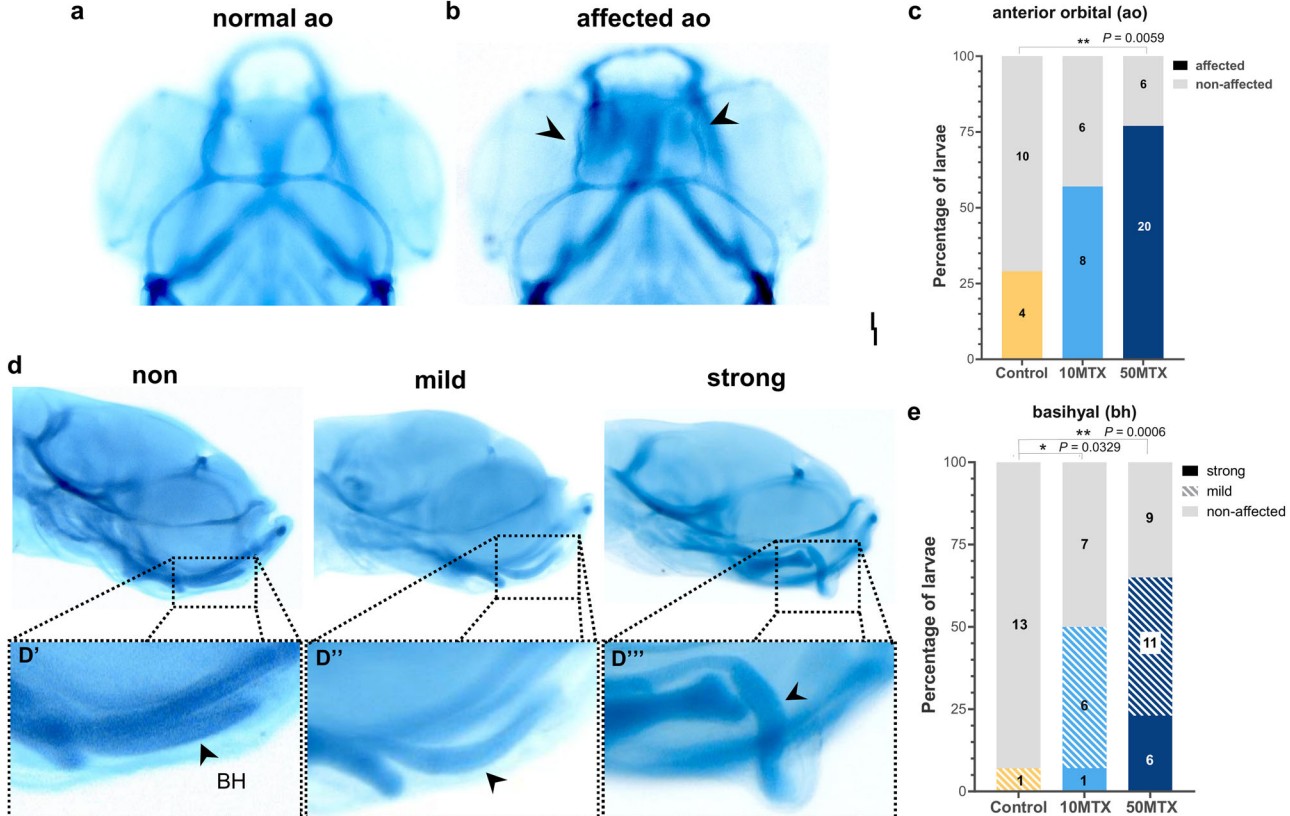

**Fig. 2 | Paternal MTX injection produced offspring's cranial cartilages malformations. a, b** Dorsal view of normal and affected anterior orbital (ao) showing a meandering shape mostly observed on the offspring from MTX treated males. **c** Quantification of the percentage of embryos presenting affected or normal anterior orbital cartilages. Numbers in the graph represent the analyzed embryos. **d** Lateral view of larvae presenting normal (trowel shape), mild (bended shape), and strong (hook shape) deformities of the basihyal cartilage. **e** Quantification of the percentage of embryos presenting normal (non-affected), mild, or strong basihyal cartilage abnormalities observed on the offspring from MTX-treated males. Numbers in the graph represent the analyzed embryos. Statistical analyses were performed using a contingency table followed by two-sided Fisher's exact test and each treatment were compared with the control independently. *P = 0.0329, **P = 0.0059, ***P = 0.0006.

Given the observed differences for certain RNA modifications, we examined the expression of specific RNA methyltransferases on the testis of control and 10MTX treated males (Fig. 5d). In agreement with our results, there was no significant change in the expression of *Trmt6* which catalyzes m1A methylation. Conversely, the expression of the enzymes that catalyzed m5C was only significantly upregulated for *Dnmt2* (p = 0.01), but not for *Nsun2* (p = 0.46). These results suggest that an increase of RNA methyltransferase expression leads to changes in the methylation status of sperm tRNAs upon MTX treatment.

### Zygotic RNA injection derived from MTX-treated sperm partially recapitulates the craniofacial phenotype

To uncover the potential role of altered small RNAs on the sperm as the causes of offspring craniofacial defects, we isolated both 20–50nt and 50–90nt RNA fractions from MTX and control (DMSO) treated males and injected them independently or combined (20–90 nt) into fertilized wild-type eggs. Initially, we did not observe differences in the percentage of hatching and embryo survival between zygotic RNA injections derived from both control and MTX-treated sperm (Fig. S4).

We have focused our analysis on the ceratohyal and basihyal cartilages because they were most affected in our initial experiments. Importantly, we found that injection of 20–50 nt, 50–90 nt or the combination of both (20–90 nt) at the two doses of MTX-treated sperm showed a significant reduction (p < 0.0001) on the ceratohyal lengths compared to control (Fig. 6a). On the other side, when we analyze the basihyal phenotypes we were unable to evidence any hook shape malformation, but instead we evidenced embryos having curved

trowel shape (affected) with the tip of the cartilage upward and downwards (Fig. 6b). Importantly, injection of 20–50 nt fraction from both 10MTX and 50MTX, but not the 50–90 nt, significantly increase the number of larvae having affected basyhial shape (Fig. 6b). Similarly, injection of 20–90 nt RNA fractions have a similar effect than the 20–50 nt. All these together suggest that RNAs from exposed males have the ability to alter the development of specific cranial cartilages on the offspring.

## Discussion

MTX binds to and inhibits dihydrofolate reductase ~~activity~~, preventing folic acid from performing its biological tasks. For more than 30 years, this drug has been used to treat immunological illnesses (including rheumatoid arthritis), blood and solid organ cancers, dermatologic diseases, and for pregnancy termination[55,56]. Despite the drug's contraindication for pregnant women due to the risk of miscarriage and birth abnormalities, the paternal influence of MTX on their offspring was largely unknown. In addition to this, the vast majority of studies in fish models such as medaka and zebrafish has been performed during embryological stages[57–60], while few have evaluated the effect on adults[61] and the consequences on their offspring.

In agreement with other studies in mice[62,63], we found that paternal MTX treatments had no effect on the fertility or survival of their progeny during the early embryonic stages. Lifetime exposure to folic acid-deficient diets, on the other hand, lead to lower sperm counts, negative consequences in progeny, and epigenetic changes[62,63]. However, this may be due to folate deficiency during

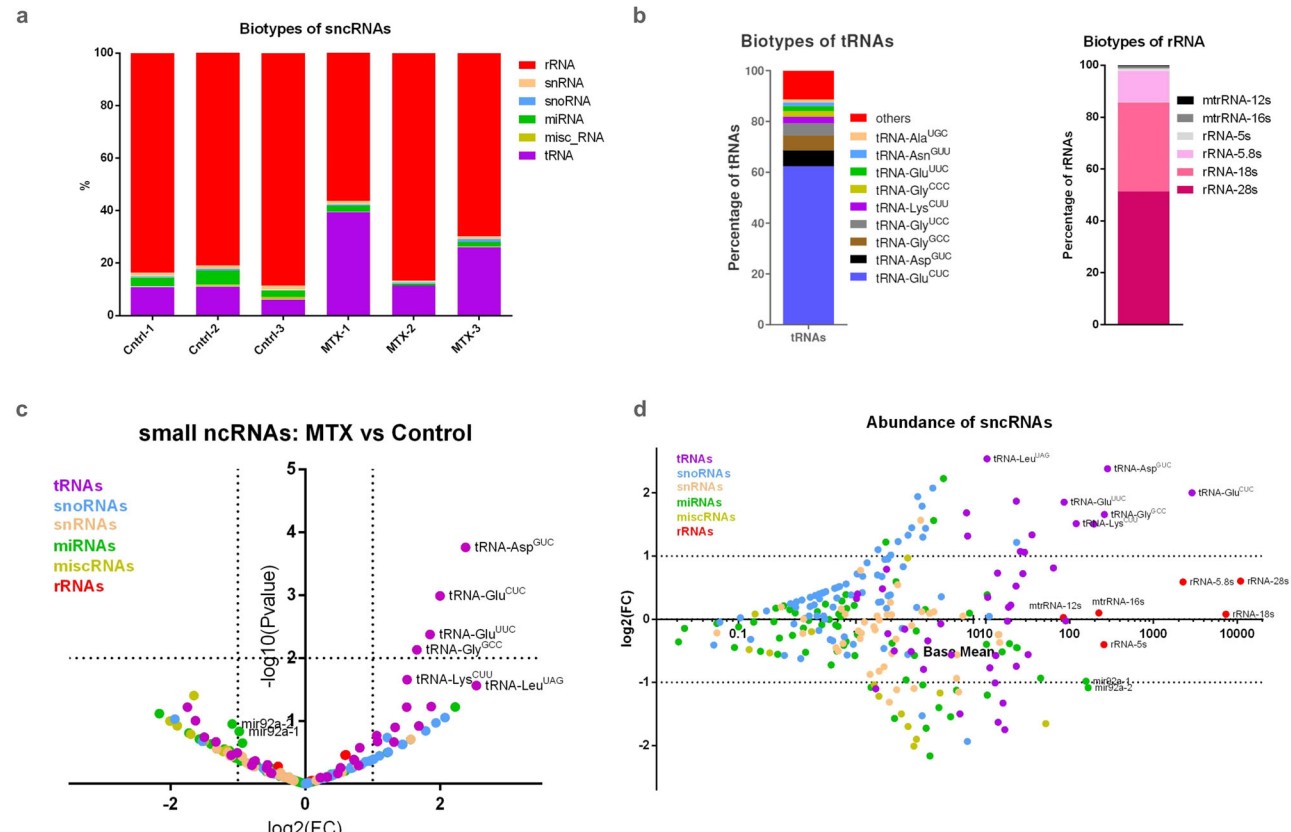

**Fig. 3 | Paternal MTX injection alters sperm sncRNAs. a** Histogram representing the comparison from sperm sncRNA on control (Cntrl1-3) and 10MTX (MTX-1-3) treated males. **b** Histogram displaying biotypes of tRNAs and rRNAs from sperm of MTX treated males. See also Supplementary Data 1 for **a** and **b**. **c** Volcano plot of depicting the fold changes in sperm sncRNAs identified as being differentially expressed within control versus MTX-treated males. Analysis of differential expression was performed by using DESeq2 which use the two-tailed Wald test. **d** MA plot displaying normalized counts (base mean) for different sncRNAs. Dotted lines depict thresholds values for significantly up and down-regulated ($\pm 1$ log$_2$ fold change and -log$_{10}$ $P$ value $\geq 1.3$). See also Supplementary Data 2 for **c** and **d**.

embryonic and post-embryonic development, which could compromise early germ cell formation and adult spermatogenesis. Moreover, major epigenetic reprogramming occurs at these periods, and multiple imprinting areas may be altered as a result of the prolonged folate shortage.

For many years, paternal contribution to offspring's health was thought to be restricted to the haploid genome of spermatozoa, whereas mother health and nutrition were linked to offspring's wellness. However, multiple recent studies have revealed that spermatozoa carry a variety of RNAs[18,64–66] capable to transmit paternal experiences[19,22–24,32]. In this regard, our work illustrated the critical significance of MTX therapy and its impact on sperm small non-coding RNA content as a possible mechanism underlying the observed craniofacial abnormalities or possibly other unexplored effects of this treatment. We discovered that tsRNAs and miRNAs are the most common small non-coding RNA in medaka sperm, which is consistent with past findings in mammals[19,30,31,66,67]. Furthermore, we revealed that tsRNAs halves changed significantly owing MTX treatment, which is in agreement with previous studies showing that tsRNAs are a dynamic population that responds to a variety of environmental stressors[19,24,68]. Particularly, we observed a higher abundance of certain 5′tsRNAs, where 5′tsRNA-Asp$^{GUC}$ was the most abundant. This result is in concordance with several studies where external factors also modulated the abundance of 5′tsRNA-Asp$^{GUC}$[20,24,31], thus highlighting the idea that certain tRNAs may be preferentially cleaved and their 5′ halves have a longer half-life compared to their respective 3′ halves.

tsRNAs can be generated through multistep cleavages, through the formation of various intermediates. Moreover, there is growing evidence that regulatory factors, such as RNA modifications and specific RNases, have a role in their specific cleavage and stability[66]. Interestingly, we found that 5′tsRNAs from Asp$^{GUC}$ and Gly$^{GCC}$ are consistently longer (-35 nt) on the sperm of MTX-treated males. This is in agreement with the discovery that small 15–22 nucleotide long fragments are normally formed in multiple tissues and cell lines[49], whereas longer 31–40 nucleotide tRNA halves are preferentially cleaved in response to different stresses[50,51].

The tsRNA functions are very speculative, but have been associated to translation, ribosome biogenesis, retrotransposition, cell-cell communication, and epigenetic inheritance, as well as how tsRNA dysregulation are related to a variety of human disorders (summarized in recent reviews[69,70]). Importantly, both tsRNAs and their precursor tRNAs are heavily modified, which contributes to multiple aspects of their function, biogenesis, stability, amino acid charging, and translational accuracy[71,72]. Our initial hypothesis was that MTX treatment may reduce the tRNA methylation thus inducing their cleavage. This is based on previous studies where the addition of m5C, which is controlled by DNMT2 and NSUN2, increase tRNA stability in flies and mice, but its deletion makes them more likely to be cleaved into tsRNAs under stress conditions[19,36,37]. However, and contrarily to our predictions, we observed that tRNAs-enriched samples (-50–70 nt) derived from MTX-treated sperm showed significantly greater levels of methylation in m5C, m2G, m7G and m2′2G. Increased levels of m5C and m2G have been observed in the 30–40 nucleotide fraction of sperm RNAs (predominantly tsRNAs) in mice fed with high-fat diets compared with those from males fed with normal diets[24]. However, it is important to mention that in our MTX-treatment we fail to observe

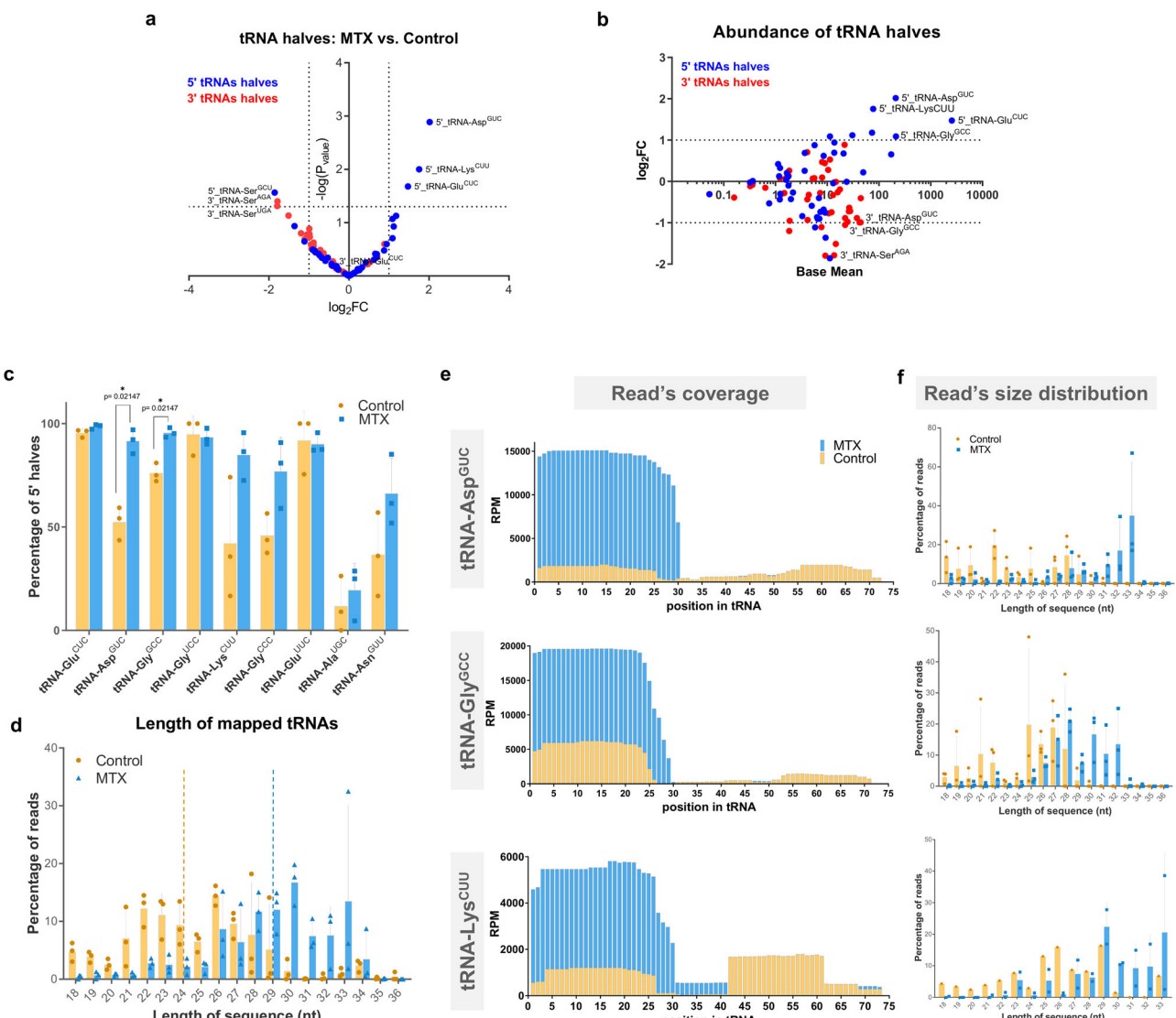

**Fig. 4 | Paternal MTX injection induced the expression and cleavage of specific tRNAs. a** Volcano plot of depicting the fold changes in sperm 5′ and 3′ tRNA halves as being differentially expressed within control versus MTX treated males. Analysis of differential expression was performed by using DESeq2 which use the two-tailed Wald test. **b** MA plot displaying normalized counts (base mean) for different 5′ and 3′tRNA halves. Dotted lines depict thresholds values for significantly up and down-regulated (±1 log₂ fold change and -log₁₀Pvalue ≥ 1.3). See also Supplementary Data 3 for **a** and **b**. **c** Histogram displaying percentage of 5′ halves relative to their corresponding 3′ halves from different tRNAs affected by MTX treatment. Asterisk indicated significant differences analyzed by multiple unpaired t-student' test followed by a correction for multiple comparison (Holm-Sidak method, with alpha = 0.05)). **d** Histogram showing the length variation of mapped tRNA reads on control and MTX-treated males. Dotted lines represent the median length. Histogram showing the read coverage (**e**) and size (**f**) distribution for the most abundant and having a significant increase in the 5′tsRNA (tRNA-Glu$^{CUC}$, tRNA-Asp$^{GUC}$, and tRNA-Gly$^{GCC}$) between control and MTX. Data on **c**, **d** and **f** represent three biologically independent replicates (*n* = 3) composed by a pool of 9 males' sperm. Values are means ± SD. See also Supplementary Data 4 for **c–f**.

differences in the methylation levels from the tsRNAs/miRNAs fraction (~20–50 nt). We speculate that because the most abundant tRNA modifications found in our study (5mC) are stated to be positioned at the 3′ end of tRNAs (positions 38C, 48C, 49C, 50C)[73], then the cleaved 3′tsRNAs halves, which accumulate the bulk of these methylations, may be preferentially degraded.

The high levels of 5mC in our 50–90nt fraction from MTX-treated males correlated with the higher expression of *Dnmt2* (also known as *Trdmt1*), but not *Nsun2*. *Dnmt2* is structurally close to other DNA methyltransferases but rather methylates only one tRNA, specifically at the cytosine 38 in the anticodon loop of aspartic acid (tRNA-Asp)[74]. The role of *Dnmt2* in paternal non-genetic inheritance have been demonstrated in mice[19,64], and recently associated with the Intergenerational effect of immune priming in insects, suggesting an evolutionary conservation of its functionality[75]. Interestingly, here we have found that

5′tsRNA-Asp$^{GUC}$ was the second most abundant tRNA in MTX-treated males and presenting a significant increase respect to their 3′tsRNA-Asp$^{GUC}$ half. In contradiction to our finding, Schaefe et al. [37]. demonstrated that m5C modification mediated by DNMT2 improves tRNA stability, where tRNA-Asp is protected from angiogenin cleavage during the heat shock response in Drosophila. In mammals, it is well known that angiogenin activity, RNase that cleaves tRNAs, is also inhibited by the presence of 5mC[19,36]. However, it is important to mention that since the endonuclease targeting the anti-codon loop of Drosophila tRNAs has not been identified yet, the authors analyzed the cleavage of tRNA-Asp induced by the addition of recombinant human angiogenin into Drosophila S2 cells[37], which may not reflect the truly physiological condition. Moreover, it has been shown that the presence of angiogenin is not mandatory for the generation of tsRNAs, and other RNases (Dicer, RNase T2, L) can also cleave tRNAs[69,76,77]. In that sense, fish does

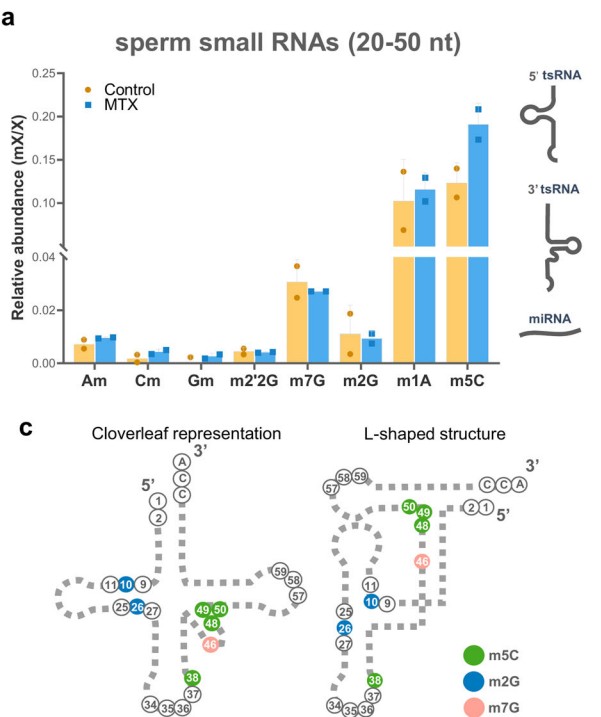

**Fig. 5 | Paternal MTX injection alters smallRNA modifications in sperm tRNA fraction and the testicular expression of RNA-methyltransferases.** Histogram comparing sperm RNA methylations on control and MTX analyzed by UHPLC-MS-MS in 50–90 nt (**a**) and 20–50 nt (**b**) fractions. Data represent two biologically independent replicates (*n* = 2) composed by RNA isolated from 9 males' sperm. Statistical analysis was performed by two-tailed unpaired t-student' test. Values are mean ± S.D. (**c**) Schematic representation of modified nucleotides in the tRNA at secondary and tertiary structure. **d** RT-qPCR for methyltransferases of m1A (TRMT6) and m5C (DNMT2 and NSUM2) on testis from control and MTX treated males, gene expression was normalized using Rpl7 and Ef1 as housekeeping genes. Statistical analysis was performed by using the two-tailed unpaired *t*-student' test. Each dot represents a biologically independent sample composed by RNA isolated from individual male testis. Values are means ± SEM.

not have angiogenin, but instead orthologues genes with the capacity to cleave tRNAs have been found[78–80] suggesting that the generation of tRNA fragments is an evolutive response against environmental stressors. In addition, it is important to mention that the activity[78], structure[79–81], and targeted dinucleotides for cleavage are different in between fish and mammals RNases[81]. These facts suggest that the overall generation of tRNA fragments is an ancient response where RNases have maintained their main role and have evolved as the organisms did it. On the other hand, the presence of 5mC, and/or other modifications, might affect their activity in a different way as it was speculated by Barraud and Tisné[82]. These authors stated that tRNA modifications are critical features of the cellular stress responses, and described the existence of a streaky crosstalk among them regulating the tRNAs stability[82]. As a result, modifications may act as a "barcode" to regulate the specific tRNA cleavage and stability resulting in the accumulation of specific tsRNAs in the sperm, which could affect the phenotype of their offspring. Finally, we showed that the zygotic injection of small RNAs isolated from MTX-treated sperm can partially reproduce the basihyal malformation and ceratohyal´s length reduction, which were also the most affected cartilages observed in the offspring of MTX-treated males. It is important to mention that the ceratohyal and basihyal cartilages form the ventral region of the hyoid arch required for the stabilization of the jaw[83] which has evolved in mammals as a structure required for milk suckling[84]. Facial characteristics and growth deficiencies have been extensively linked to both folate deficiency and fetal alcohol syndrome. This phenotypic linkage is, in part, because chronic alcohol abuse affects the folate levels by reducing their initial hydrolysis and subsequent uptake into the cells[85]. Furthermore, fetal alcohol syndrome has also been connected with a weak sucking ability and other feeding difficulties in humans[86], which may presume a hyoid malformation.

In summary, our data suggests that paternal MTX-exposure influenced sperm tRNA methylation, as a result of alterations in the expression of certain RNA methyltransferases. These epitranscriptomic changes may cause the selective tRNA cleavage and the maintenance of certain 5' tRNA halves. These changes in the sperm RNA content and modifications might affect transcriptional cascades in the fertilized oocyte, with possible implications in cranial cartilage formation (see hypothetical model in Fig. 6c). The understanding of how tRNA modifications and their derived fragments impact on the transcriptional cascades occurring during early embryo will provide valuable insights into several diseases and it is expected that this will be a main focus of research in this field in the near future.

## Methods
### Medaka Husbandry
All experiments were performed with medaka fish (*Oryzias latipes*) (strain hi-medaka, ID:MT835) supplied by the National BioResource Project (NBRP) Medaka (http://www.shingen.nig.ac.jp/medaka/). Fish were maintained and fed following standard protocols for medaka[87]. Fish were handled on the Care and Management of Laboratory Animals (http://www.ufaw.org.uk) and internal approved regulations (SICUAE-University of San Martín 33/2022). Adult fishes were divided and acclimatized in 4L fish tank during 3 weeks under a constant photoperiod (14L:10D) and controlled temperature (26 ± 0.5 °C), prior to experimental procedures.

### Experimental design
Adults medaka fish were divided into 3 groups composed by 3 independent replicates having 9 males and 2 females per tank (all of them having a body weight of ~200 mg). After the acclimatizing period, each

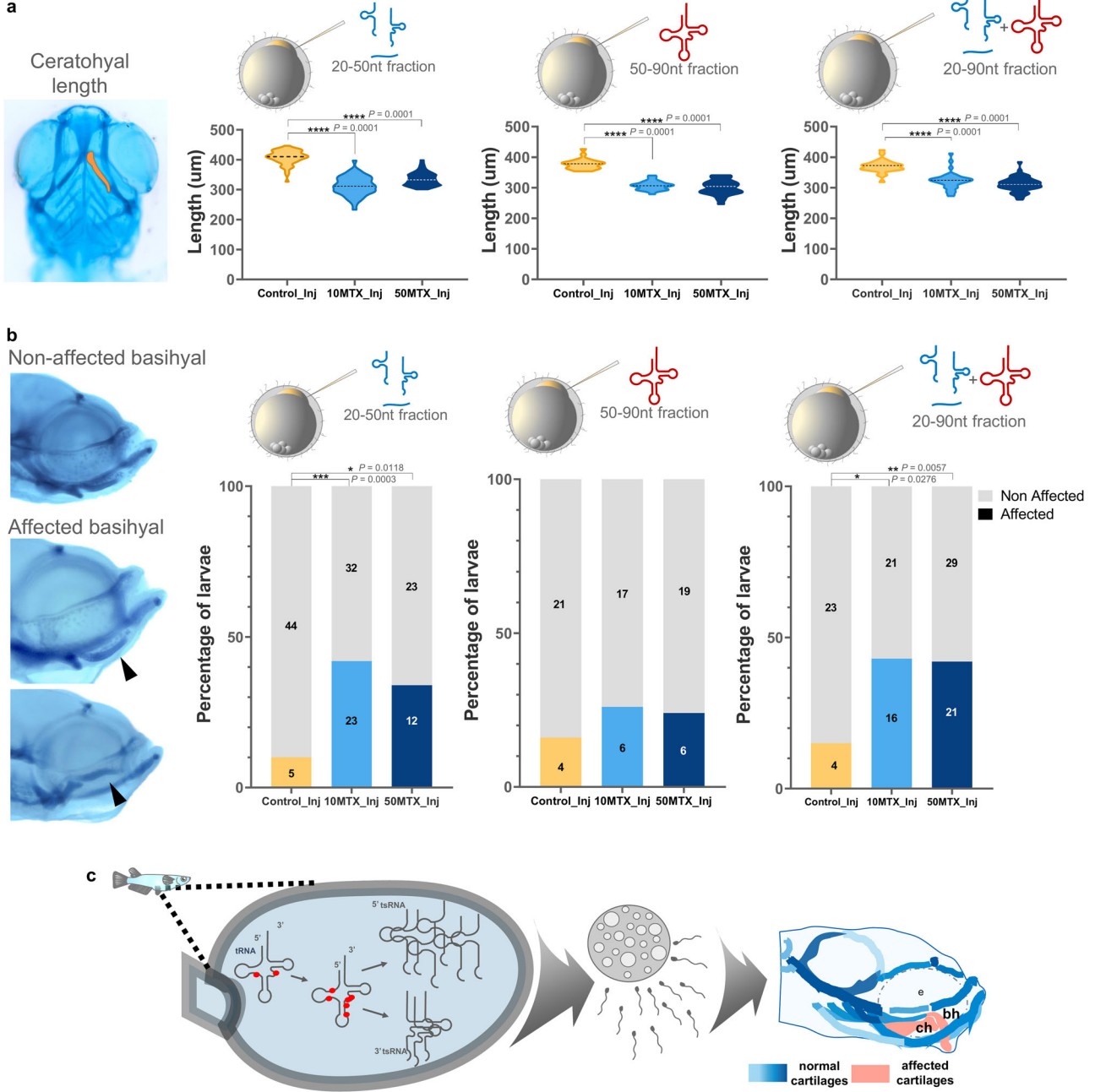

**Fig. 6 | Zygotic RNA injection derived from MTX-treated sperm partially reca-pitulates the craniofacial phenotype. a** Violin plots represent the measurement of ceratohyal lengths on wild-type fertilized eggs injected with sperm-RNA frac-tions (20–50nt, 50–90nt, or both together) obtained from control, 10MTX and 50MTX treated males. Statistical analyses were performed using ANOVA one-way followed by multiple comparison Tukey's test. **b** Lateral view of larvae presenting non-affected (trowel shape) or affected (bended upward or downward) basihyal cartilage shape. Quantification of the percentage of embryos presenting affected or non-affected basihyal cartilage abnormalities from zygotes injected with sperm-RNA fractions (20–50nt, 50–90nt, or both together) obtained from control, 10MTX and 50MTX treated males. Numbers in the graph represent the analyzed embryos. Statistical analyses were performed using a contingency table followed by two-sided Fisher's exact test. **c** Proposed model summarizing the results.

male was intraperitoneal injected with control solution (PBS/1% DMSO), 10 mg of MTX per kg of body weight (10 mg/kg MTX) or 50 mg/kg MTX (A6770-SigmaAldrich, diluted in PBS/1%DMSO). Briefly, males were anesthetized with 1% benzocaine solution (Parafarm), gently dried with a paper towel, and placed in a dampened sponge ventral side up, with their anal fin and cloaca exposed. Immediately, using a 10ul syringe (Hamilton), fish were injected using a binocular stereoscope (Nikon SMZ745) and then returned to their tanks for 7 days until sperm collection for in vitro fertilization and small RNA extraction.

## In vitro fertilization

Sperm collection was carried by anesthetizing the fish and placed in a dampened sponge ventral side up following published protocols for medaka[87]. A micro-forceps was used to gently strip the fish and the released semen was collected by using a micropipette (~0.5 μl/fish) and pooled for the posterior in vitro fertilization and small RNA extraction. For the in vitro fertilization, 0.2 μl from obtained sperm were used to fertilize a pool of 24–28 eggs collected from mature untreated females. Fertilized eggs were immediately transferred and incubated in 60 mm petri dishes with embryo media (17 mM NaCl, 0.4 mM KCl, 0.27 mM

CaCl$_2$.2H$_2$O, and 0.66 mM MgSO$_4$; pH:7) until 3 days' post hatching (dph). Incubation was monitored and the percentage of fertilization and survival until hatching was evaluated.

## Alcian blue staining

Cartilages from embryos were analyzed at 3 dph by using alcian blue staining. Larvae were fixed in 4% paraformaldehyde overnight at 4 °C and washed three times with PBSw (PBS-0.1% tween20). After that, embryos were incubated in a bleaching solution (0.5X SSC, 5% formamide, 10% hydrogen peroxide) and exposed to light during 2 h. Larvae were washed several times with PBSw and immediately incubated in alcian Blue solution (0.1%p/v alcian blue, 0.37%v/v HCl, 70%v/v EtOH) for 1 h on a nutator. Then, larvae were washed five times with 01%v/v HCl-70%v/v EtOH for 30 min on a nutator; the last wash was left overnight at room temperature. Next, larvae were washed six times with 50%v/v glycerol-0.5%v/v KOH for 30 min on a nutator and the last wash was left overnight. Finally, larvae were washed four times with the same solution and left in 90% glycerol-10% ETOH for imaging processing and phenotype analysis. Larvae were photographed at ventral, dorsal and lateral view by using a trinocular stereoscope (SteREO Discovery v20. Zeiss) and analyzed using the ImageJ software[88].

## Small RNA extraction and library preparation

Small RNAs were isolated from sperm following manufacturer's instructions (illustra RNAspin Mini RNA isolation kit-GE Healthcare). The 3′ adapters (see Supplementary Table 2 for full list of adapter oligos utilized for library preparation) were ligated using SRBC barcode adapters for each sample, additionally 18-mer and 30-mer markers were ligated and used as control for the ligation process and markers for the product correct size. The 3′-ligated small RNAs were size selected using 15% denaturing urea polyacrylamide gels at a constant power of 40–50 W for ~30 min and stained by using SYBR Gold 0.05%V/V in TBE 0.5X and the 3′ ligated RNAs ranging from 18 −30 mer were cut out. RNAs were purified using Zymo PAGE elution kit (ZR™ small RNA PAGE recovery kit) according to manufacturer's instructions, the elute 3′-ligated small RNAs were elute in 5′ linker mix containing 5′ adaptor. The 3′-ligated RNAs + 5′ adapter were denaturated for 5 min at 70 °C, cooled on ice immediately, ligated with T4 RNA ligase (NEB) and incubated at 16 degrees overnight. Ligated small RNAs were purified by using MBS beads, briefly: MBS buffer, MBS bead slurry (beads + buffer), mixed by vortexing, added isopropanol and incubated at room temperature. Beads were separated on magnet and the supernatant was removed, after several washes with ethanol the RNA was eluted with ultrapure water and transferred into PCR strip. For reverse transcription, small-RNAseq RT primer to each sample were used and a negative control without reverse transcriptase was included, Superscript II reverse transcriptase was used to obtain the synthesis of the first strand. To amplify cDNA libraries, KAPA HiFi Real Time Library Amplification Kit (Roche) was used; PCR were performed using TruSeq Universal Adapter primer (Solexa_PCR_fwd) and TruSeq Index reverse primers (Solexa_IDX_rev), this latter includes barcodes assigned to each different sample. Briefly: master mix was added and TruSeq Index reverse primer were added to PCR strips containing cDNAs; then KAPA HiFi HS RM and Truseq Universal Adapter primer were added to the mix. The cycling program was: Denaturation at 98 °C for 45 s; 20 cycles of 98 °C for 15 s, 65 °C for 30 s, 72 °C for 30 s, 72 °C for 10 s; and a final extension at 72 °C for 1 min. The amplified cDNA was purified by using 3% Low-Range Ultra Agarose gel (Bio-Rad) according to the manufacturer's instructions at constant 80–100 V using GeneRuler 50 bp DNA Ladder (ThermoFischer Scientific) as molecular marker. Gel was visualized on a long wave UV transilluminator and DNA band between 150–200 bp were excised using a clean scalpel blade and put into a clean 15 ml Falcon tube; the DNA was purified using the Zymoclean Gel DNA recovery kit (Zymo Research) according to manufacturer's instructions.

## Bioinformatics analysis

Adapters from reads were removed using CUTADAPT, the output were reads ≥15 bp, reads whose adapters were not identified were discarded. The output of 15 bp were used to analyze differential expression of sncRNAs (miRNAs, tRNAs, snRNAs, snoRNAs, and rRNAs) and differential expression of tRNAs, 5′ tRNA halves and 3′ tRNA halves by different strategies. First, differential expression of sncRNAs was analyzed on reads where the random nucleotides on 5′ (4 bp) and 3′ (6pb) were cut using FASTQ Trimmer. The obtained reads having <19 bp were discarded using Filter Fastq and the remaining reads were aligned against the medaka genome (Assembly ASM00223467v1) with RNA STAR (allowing multimapping reads, 1 mismatch, and not allowing introns). Expression of miRNAs, tRNAs, snRNAs, snoRNAs was analyzed using FeatureCounts (allowing multimapping reads to be counted, and assigning 1/n fractions to multimapping reads) with Ensembl annotation (Release v102). To analyze cytoplasmic and mitochondrial rRNA expression, reads were mapped to a custom fasta file containing 28s rRNA (RNA central: URS000215D18B_8090), 18s rRNA (refseq: XR_002874070.1), 5.8s rRNA (RNA central: URS0000671FD1_8090), 5s rRNA (refseq: XR_002875036.1), 16s mtRNA (RNA central: URS00003A7D46_8090) and 12s mtRNA (RNA central: URS000033338A_8090) sequences. Differential expression of all sncRNAs was calculated using DESEQ2. Second, to analyze differential expression of 5′ and 3′ tRNA halves, an additional 3 base pairs were removed with FASTQ Trimmer from the 3′end of all reads. Reads having less than 15 bp were discarded using Filter Fastq. The output was aligned to the reference genome and analyzed as mentioned before using custom GTF files with genomic coordinates for either 5′ or 3′ tRNA halves. To determine the sequence length of mapped tRNA and rRNA reads, BAM files were filtered (using GTFs files containing genomic coordinate for full length tRNAs o rRNAs), reads were extracted, converted to fasta and their length computed with in-house scripts. tRNA and rRNA read coverage was calculated with Bam-Coverage (bin size 1, no smoothing) with RPM values representing reads per million mapped to functional sncRNA categories (miRNAs, tRNAs, snRNAs, snoRNAs and rRNAs).

## UHPLC-MS-MS

The analysis of modified ribonucleotides from spermatic RNAs were performed by UHPLC-MS-MS. For that purpose, ~1.5 μg of total RNAs were isolated from two independent pools of stripped sperm from ~9 control and 10MTX-treated males and run in denaturant polyacrylamide gel (15%, 7 M Urea). The gel was then stained with ethidium bromide and RNAs that have a ranged size from 20–50 nt and 50–90 nt were cut and recovered using the ZR small-RNA™ PAGE Recovery Kit (Zymo Research) by following the manufacturer's instruction. Approximately 100 ng of RNA was obtained on each fraction and utilized for UHPLC-MS-MS analysis. Then, 100 ng purified RNA samples were digested to nucleosides for 2 hr at 37 °C using the Nucleoside Digestion mix (NEB, M069S). Quantifications were performed as in[89], briefly: digested RNA samples were diluted to 100 μl with ddH20 and filtered through 0.22 μm Millex Syringe Filters. 5 μl of the filtered solution was injected for LC-MS/MS analysis using the Agilent 1290 UHPLC-MS/MS system with a Hypersil Gold C18 reversed-phase column (2.1 × 150 mm, 3 μm). Mobile phase A consisted of water with 0.1% (v/v) formic acid and mobile phase B consisted of acetonitrile with 0.1% (v/v) formic acid. Mass spectrometry detection was performed using an Agilent 6470 triple quadrupole mass spectrometer in positive electrospray ionization mode and data were quantified in dynamic multiple reaction monitoring (dMRM) mode, by monitoring the mass transitions 268↠136 for Adenosine (A), 282↠150 for N6-methyladenosine (m1A), 244↠112 for Cytidine (C), 258↠126 for C5-methylcytidine (m5C), 284↠152 for Guanosine (G), 298↠166 for N7′-methyladenosine (m7G) and N2-methylguanosine (m2G), 312↠180 for N2,N2-dimethylguanosine (m2′2G), 282↠136 for 2′-O-

methyladenosine (Am), 258☞112 for 2′-O-methylcytidine (Cm) and 298☞152 for 2′-O-methylguanosine (Gm). To quantify the concentrations of the various methylation modifications we used pure nucleosides of A, C, G, m1A, m5C, m7G, m2G m2'2G, Am, Cm, and Gm to generate calibration standard curves through serial dilution.

## RNA quantification by RT-PCR

Dissected testis from adult males (Control and 10MTX groups) were used for gene expression analysis. For this purpose, a complete testis from 8 males were individually grinded in 300 μl of TRIzol Reagent (Life Technologies) and total RNAs (~800 ng/testis) were isolated and retrotranscribed[90]. Expression of target genes were measured by qPCR using Fast Start Universal SYBR green Supermix (Roche Diagnostics, USA) on Thermal Cycler StepOne Plus (Applied Biosystem, USA), using ribosomal protein L7 (rpl7) and elongation factor 1 alpha (ef1α) as reference genes with the geometric mean calculation as described by Padilla et al. [90]. Real-time PCR primers are listed in Supplementary Table 1. Each sample was run in duplicate and a PCR reaction without the addition of template, was used as negative control. The amplification protocol consisted of an initial cycle of 1 min at 95 °C, followed by 40 cycles: 15 s at 95 °C and 30 s at 60 °C. After the amplification, a melt curve was performed by 1 cycle: 15 s at 95 °C, 60 s at 60 °C, and 15 s at 95 °C enabling confirmation of amplification of single products. Gene expression levels were calculated by the $2^{-\Delta\Delta Ct}$ comparative threshold cycle (Ct) method (where $\Delta\Delta Ct = \Delta Ct$ sample - $\Delta Ct$ reference). The efficiency of amplification ranged 95–105% for all genes studied. The expression level in each group was normalized to the control and was presented as a fold of change[91].

## Zygotic RNA injection

RNAs were isolated from pools from stripped sperm from ~9 control, 10MTX, and 50MTX-treated males and run in denaturant polyacrylamide gel (15%, 7 M Urea). RNAs that have a ranged size from 20–50 nt and 50–90 nt were cut and recovered using the ZR small-RNA™ PAGE Recovery Kit (Zymo Research) by following the manufacturer's instruction. Fertilized eggs at the stage of one cell were injected with 45.1 pg of purified RNAs from 20–50 nt, 50–90 nt or a combination of both (20–90 nt) by using the microinjector Nanoject II (Drummond Scientific Company). After this, the injected embryos were transferred into petri dish containing embryo media and let them grown up until 3 days post hatching. The embryos were monitored every day and alcian blue staining was performed for larvae as previously mentioned. Stained larvae were analyzed as previously mentioned.

## Reporting summary

Further information on research design is available in the Nature Portfolio Reporting Summary linked to this article.

# Data availability

Trimmed sequencing data that was generated in this study for the initial bioinformatics analysis have been submitted to the NCBI Sequence Read Archive under BioProject ID PRJNA857097. LC-MS/MS data have been uploaded to Metabolights [http://www.ebi.ac.uk/metabolights/MTBLS7368]. Source data are provided with this paper.

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

## Acknowledgements

We thank all the authors and members in LBD for their contribution and helpful discussions during the course of our study. This work was supported by the Agencia Nacional de Promoción Científica y Tecnológica (PICT 2018-1879 to P.H.S-M.). J.P.T. would like to acknowledge support from Universidad de la República, Uruguay (CSIC I+D_2020_433). Work in the Greer lab was supported by an NIH grant (DP2AG055947) to E.L.G.

## Author contributions

N.A.J. and P.H.S-M. designed, performed the experiments, and wrote the manuscript; L.C. and E.M.S. contributed to making the sRNAs libraries and sequencing; J.P.T and M.C. contributed on the bioinformatics analysis of small RNAs; E.L.G. and K.B. contributed on the RNA methylation analysis; J.I.F., A.B., and L.A.P. contributed on the fish handling, sperm acquisition, eggs fertilization and RT-qPCRs. All the authors contributed on the final manuscript edition.

## Competing interests

The authors declare no competing interests.
