## [Peer Review File · Nature Communications]

Paternal methotrexate exposure affects sperm small RNA content and causes craniofacial defects in the offspringReviewer #1 (Remarks to the Author):

In this interesting investigation, the authors reported the effect of paternal methotrexate exposure in inducing offspring craniofacial development defects in medaka fish; as a potential mechanism, the authors report that paternal methotrexate injection alters sperm sncRNA profiles, with particular emphasis on tRNA-derived small RNAs (tsRNAs), they also found related changes in RNA modifications and testicular expression of RNA methyltransferases that may explain the observed altered biogenesis of tsRNAs. The data are potentially important to establish a relationship between sperm sncRNA & modification and paternal methotrexate induced offspring phenotypic modulation, however, an important experiment is missing regarding whether the injection of sperm RNAs from exposed father can sufficiently induce offspring phenotype. I suggest the authors to perform this essential experiment as it may provide a critical insight to the issue being explored. Please find more of my comments below:

1. First and foremost, as I mentioned in the overall comment above, it would be essential for the authors to perform an zygotic RNA injection experiments by comparing the sperm RNAs extracted from normal vs methotrexate-exposed males, this will provide more direct evidence regarding whether the observed sperm sncRNA/modifications changes are causatively related to the phenotype observed.
2. In addition to the craniofacial defects described, did the author noticed different penetrance in male and female offspring? Are there other phenotypes observed, for example, are the F1 reproductive normal? If yes, did the authors tried to trace the phenotype in F2?(just out of curiosity)
3. For the analysis of sperm RNAs, in addition to tsRNAs, did the author also analyzed rRNA-derived small RNAs (rsRNAs)? This is because rsRNAs are also increasingly recognized to be similarly sensitive to environmental exposures (Nat Cell Biol 2018 PMID: 29695786; Plos Biol 2019 PMID: 31877125). And in fact, rsRNAs turn out to be the most abundant sncRNAs in mouse sperm using recently updated RNA-seq method (Nat Cell Biol 2021 PMID: 33820973).
4. When discussing the function of Dnmt2, there is a recent publication showing that it is also involved in the Intergenerational effect of immune priming in insects, suggesting an evolutionary conservation of its functionality (Insect Mol Biol 2022, PMID: 35790040), this could be discussed to enrich the context.

Paternal methotrexate exposure affects sperm small RNA 1 content and causes craniofacial defects in the offspring

Reviewer #2 (Remarks to the Author):

The manuscript by Jimenez et al. "Paternal methotrexate exposure affects sperm small RNA content and causes craniofacial defects in the offspring" describes the effect of methotrexate (MTX) treatment of medaka sperm on sperm development and later embryonic development. The authors use a methotrexate (MTX) treatment to reduce folate levels and study the effect on male sperm and subsequent embryonic development after IVF.

The manuscript reports changes of SncRNAs in treated sperm and descriptive work showing (some) endpoints/effects on embryonic development. Attempts to elucidate the underlying mechanisms that cause these late craniofacial defects are lacking.

The authors do not show that MTX treatment results in a folate reduction in treated males. Importantly they do not connect the reported MTX effects of SncRNAs and RNA-methyltransferases in male testis with the late occurring (minor) very localized craniofacial malformations in IVF derived embryos. A more detailed study of how embryonic development is affected by MTX induced alterations of SncRNAs is lacking. Also an analysis of the paternal genome in the sperm to rule out possible genomic defects by MTX treatment is lacking. Thus the work leaves the reader with a large gap between (some) early effects of MTX treatment on sperm development and subtle, late occurring embryonic malformations. This makes it difficult to evaluate the functional relevance of the reported early effects of SncRNA content and modifications.

Specific points:

The authors should explain/justify the methotrexate treatment, why they chose a single peritoneal injection followed by a 7 day recovery phase prior to sperm collection. Is this the result of tests or based on published data that is not cited?

The authors imply that this methotrexate treatment results in a folate deficiency in the treated males. They should examine whether this is true and quantify the effect of MTX treatment on folate levels in treated males. To this end they should monitor methotrexate levels after the injection. By that, the authors could lend support to the proposed mode of action of MTX (reducing folate levels). However, this would still not rule out that methotrexate has other effects that lead to the reported effects rather than the implied effect on folate synthesis. In view of the reported effects of MTX on DNA synthesis it would be interesting to see whether the authors can rule out a possible effect on the MTX treatment on the integrity/completeness of the paternal genome. Late phenotypes as mentioned above could be indicative of such an effect.

cranial cartilage malformations:

Treated and control sperm was used for IVF of pools with 24-28 oocytes. Figure 2 C & E give some numbers of hatchlings. The authors do not report on the replication, i.e. reproducibility of the effects. For example: how many different treated and control males were used for the IVF? Was more than one IVF derived batch of developing eggs monitored?

Apart from hatching time and rate and craniofacial morphology no further description of IVF derived offspring is given. Did the authors monitor also later development of the IVF derived hatchlings? Did craniofacial malformations persist? Were other malformations or defects detectable (heart development/heart rate, growth, fertility, behaviour)?

The data on the craniofacial malformations describes a late endpoint. In view of the approximately 20 days period between MTX treatment and phenotype analysis, the many developmental processes that take place during that time (spermatogenesis, entire embryonic development), it would be interesting to know more about the (molecular genetic) cause that culminates in these effects.

Since the craniofacial malformations are very specific and localized, potential pathways (and candidate genes) should be identifiable. For example: did the authors look at neural crest development that precedes cranial cartilage formation? This could also shed light on the genes/pathways that are affected by the methotrexate induced modifications of the sperm genome and SncRNAs. See also references #58-60.

In view of established genome, transcriptome and proteome analysis techniques, it would be possible to carry out a more detailed analysis, for example of the transcriptome. This could shed light on the hypothesis that the reported alterations of specific SncRNAs cause the observed embryonic malformations.

Figure S1 D shows a delay of hatching of treated embryos. Is this significant? If yes, the authors should mention/discuss this "phenotype".

Reviewer #3 (Remarks to the Author):

This manuscript by Jimenez et al. examines the consequences for offspring of male medaka exposed to methotrexate. The effects on offspring from maternal exposure is well documented. Here, the authors show craniofacial defects occurring after males were given low and high doses of the drug. There is evidence that environmental or disease effects produce changes in small non-coding RNAs in sperm cells, which may alter developmental programs. The authors examined small RNAs and showed changes in tRNA expression and RNA modifications. Overall, this is a

strong contribution. However, the authors should make more conservative conclusions. The changes in small non-coding RNAs may or may not lead to craniofacial defects. There was no evidence presented to show any causation between the expression changes and phenotype. I have specific comments below.

1- The numbers of animal used to generate the statistics should be given in the results or figure legends.

2. The light color of the bars in Figure 4 and 5 are very hard to see. The contrast between the bar color and the white page should be changed to improve visibility.

3. Line 192. As the sentence reads, the authors give the impression that MTX binds to the enzyme activity. Removing the word "activity" would solve the issue.

4. Line 195. The sentence should read "and for pregnancy termination."

5. Line 233. "Up to now" is stated like there is a difference after the findings in this paper. The tsRNA functions in this instance remain speculative. More conservative statements are needed here and elsewhere to avoid giving the impression that the data presented here connects the gene expression changes with phenotype changes. Causation was not established. There is only a correlation, which should be stated clearly.

REVIEWER COMMENTS

Reviewer #1:

First of all, we would like to express our gratitude to reviewer 1 for the positive criticisms. All the comments are very appropriated and constructive, which is highly appreciated. All the changes in this new version of the manuscript are highlighted in yellow.

1. First and foremost, as I mentioned in the overall comment above, it would be essential for the authors to perform a zygotic RNA injection experiments by comparing the sperm RNAs extracted from normal vs methotrexate-exposed males, this will provide more direct evidence regarding whether the observed sperm sncRNA/modifications changes are causatively related to the phenotype observed.

The observation of the reviewer is valid. Accordingly, we have now isolated from MTX- and Control- treated males small-RNA fractions (20-50 nt and 50-90 nt) and injected them independently and combined (20-90 nt) into fertilized wild-type eggs.

Our results show that injection of 20-50 nt, 50-90 nt or the combination of both (20-90 nt) at the two doses of MTX-treated sperm showed a significant reduction ($p < 0.0001$) on the ceratohyal lengths compared to control (Fig. 6A). On the other side, when we analyze the basihyal phenotypes we were unable to evidence any hook shape malformation, but instead we only evidenced embryos having curved trowel shape bended upward or downward (affected) (Fig. 6B). Importantly, injection of 20-50 nt fraction from both 10MTX and 50MTX, but not the 50-90 nt, significantly increase the number of larvae having affected basihyal shape (Fig. 6B). Similarly, injection of both 20-90 nt RNA-fractions have a similar effect than the 20-50 nt. All these together suggest that RNAs from exposed males have the ability to alter the development of specific cranial cartilages on the offspring.

2. In addition to the craniofacial defects described, did the author noticed different penetrance in male and female offspring? Are there other phenotypes observed, for example, are the F1 reproductive normal? If yes, did the authors tried to trace the phenotype in F2?(just out of curiosity).

Unfortunately, we did not examine the genetic sex of the offspring to determine differential penetrance of craniofacial defects in male and female offspring. Although it would be very interesting, on this occasion we didn't grow them longer to evaluate their reproductive capacities and the penetrance of the trait in their offspring.

3. For the analysis of sperm RNAs, in addition to tsRNAs, did the author also analyzed rRNA-derived small RNAs (rsRNAs)? This is because rsRNAs are also increasingly recognized to be similarly sensitive to environmental exposures (Nat Cell Biol 2018 PMID: 29695786; Plos Biol 2019 PMID: 31877125). And in fact, rsRNAs turn out to be the most abundant sncRNAs in mouse sperm using recently updated RNA-seq method (Nat Cell Biol 2021 PMID: 33820973).

Following the reviewer suggestion, we updated the transcriptome analysis to include Medaka's rRNA-derived small RNAs (see changes to materials and methods highlighted in yellow). As the reviewer correctly points out, and in accordance with previous studies (Nat Cell Biol 2021 PMID: 33820973), small RNA-seq reads mapping rRNA are very abundant and account for 78% of sncRNAs reads (Figure 3A). These reads mostly mapped to 28S, 18S and 5.8S rRNA (Figure 3B). Reads mapping 5S rRNA and both 16S and 12S mtrRNA were also detected, albeit in a much lower proportion (Figure 3B). Despite being the most abundant sncRNAs, we found no

statistically significant difference in either cytoplasmic or mitochondrial rRNA expression between control and methotrexate-treated fish (Figure 3 C-D). Importantly, coverage plots for rRNA 28S, 18S and 5.8S showed almost identical coverage patterns in both experimental conditions, confirming that no specific rRNA fragment becomes enriched or depleted by the treatment (Figure S3).

In the first version of this manuscript, we found a minor shift towards larger rRNA fragments in methotrexate treated fish (Original Figure S2C). This analysis was performed using Medaka's reference genome (Assembly ASM00223467v1) and Ensembl annotation (Release v102) in which only 18s and 5S rRNAs are present. To better understand rRNA expression and fragment distribution, we identified 28S rRNA, 5.8S rRNA, 16S mtrRNA and 12S mtrRNA sequences (see changes y materials and methods) and performed additional analysis. Consistent with our previous results, we found that rRNA-derived fragments mapped from 28S, 18S, 5.8S, 16S and 12S are slightly larger after methotrexate treatment (Figure S3), but this is not as dramatic as observed for tsRNAs (median length of 22 nt to 24 nt for rRNAs, vs. 24 nt to 29 nt for tsRNAs; Figure S3B).

Taken together, our findings suggest that methotrexate has a transcriptional effect on sperm sncRNA, primarily affecting tRNA expression and fragmentation, most likely due to increased Dnmt2-dependent methylation, which could alter their biogenesis or subsequent stability.

4. When discussing the function of Dnmt2, there is a recent publication showing that it is also involved in the Intergenerational effect of immune priming in insects, suggesting an evolutionary conservation of its functionality (Insect Mol Biol 2022, PMID: 35790040), this could be discussed to enrich the context.

This is a great suggestion, and thanks for bringing this paper to our attention. We have now discussed about this article in the discussion.

Reviewer #2:

We would like to thanks reviewer 2 for the careful reading of our manuscript and for the suggested experiments/changes that really improve this new version. All the changes in this new version of the manuscript are highlighted in yellow.

The authors should explain/justify the methotrexate treatment, why they chose a single peritoneal injection followed by a 7 days recovery phase prior to sperm collection. Is this the result of tests or based on published data that is not cited?

Methotrexate has been used as an anti-folate in a number of experimental studies in female mammals (Mouse, rat, rabbit, cat, and monkey reviewed in Hyoun et al., 2012, DOI: 10.1002/bdra.23003), with intraperitoneal/intravenously administered doses ranging from 0.3 to 50 mg/kg causing malformations in their descendants mostly at concentrations greater than 10 mg/kg. Unfortunately, there are no publications on fish, so we based our doses on those publications made in mammals.

Dramatic remodeling of the sperm small RNA repertory has been described in mice (Dev Cell 2018, PMID: 30057276) and zebrafish (Sci Adv 2016, PMID: 27500274) that occurs during their maturation. In medaka, to go from spermatogonia to sperm takes about 7 days, and since we didn't know at which stage of sperm maturation the smallRNAs RNAs could be acquired and/or modified, we have made preliminary tests at 5, 7 and 9 days post injection (dpi). We found that at 5dpi only the higher doses (50mg/Kg) produced a penetrant craniofacial phenotype, at 7dpi

both lower and higher doses (10mg/kg and 50mg/kg) produced offspring defects, and at 9dpi none of the tested doses consistently produced offspring defects. Based on this, we give 7 days post-injection to do our posterior analysis.

The authors imply that this methotrexate treatment results in a folate deficiency in the treated males. They should examine whether this is true and quantify the effect of MTX treatment on folate levels in treated males. To this end they should monitor methotrexate levels after the injection. By that, the authors could lend support to the proposed mode of action of MTX (reducing folate levels). However, this would still not rule out that methotrexate has other effects that lead to the reported effects rather than the implied effect on folate synthesis. In view of the reported effects of MTX on DNA synthesis it would be interesting to see whether the authors can rule out a possible effect on the MTX treatment on the integrity/completeness of the paternal genome. Late phenotypes as mentioned above could be indicative of such an effect.

Unfortunately, we are unable to measure folate levels in our lab. Although we cannot rule out the possibility that MTX affects the paternal genome integrity at some level, the fact that paternal MTX treatments had no effect on the fertility or survival of their progeny during the early embryonic stages argues against this possibility. Moreover, if that is the case, we would expect a more pleiotropic effect than a specific phenotype in craniofacial development. Although we are unable to measure folate levels, we still consider that since methotrexate has been used clinically in the treatment of malignancy, psoriasis, rheumatoid arthritis, and other autoimmune and inflammatory disorders, our findings may provide evidence for future medical care recommendations for males taking MTX while trying to conceive, which have never been elucidated.

Cranial cartilage malformations:

Treated and control sperm was used for IVF of pools with 24-28 oocytes. Figure 2 C & E give some numbers of hatchlings. The authors do not report on the replication, i.e. reproducibility of the effects. For example: how many different treated and control males were used for the IVF? Was more than one IVF derived batch of developing eggs monitored?

Thanks for pointing this out. We have now clarified in the M&M section that we used 3 independent pools of sperm from 9 males reared in different aquaria to fertilize groups of 24-28 oocytes.

Apart from hatching time and rate and craniofacial morphology no further description of IVF derived offspring is given. Did the authors monitor also later development of the IVF derived hatchlings? Did craniofacial malformations persist? Were other malformations or defects detectable (heart development/heart rate, growth, fertility, behaviour)?

In our experiments, we monitored the percentage of fertilization and hatching, the hatching time, and the survival curve of the embryos until day 10 post-hatching (Fig. S1). None of those parameters evidenced significant differences between control and MTX (10 and 50mg/kg). We have also monitored the embryos for general malformations (not detected in our embryos), and specifically, we only focused on craniofacial malformations at the analyzed time point. We agree with the reviewer that analyzing heart development and behavior would be also very interesting to address in further studies.

The data on the craniofacial malformations describes a late endpoint. In view of the approximately 20 days period between MTX treatment and phenotype analysis, the many developmental processes that take place during that time (spermatogenesis, entire embryonic development), it would be interesting to know more about the (molecular genetic) cause that culminates in these effects.

Since the craniofacial malformations are very specific and localized, potential pathways (and candidate genes) should be identifiable. For example: did the authors look at neural crest development that precedes cranial cartilage formation? This could also shed light on the genes/pathways that are affected by the methotrexate induced modifications of the spermgenome and SncRNAs. See also references #58-60.

Only a fraction of the cranial migratory neural crest cells will contribute to the basihyal and ceratohyal, which are the most affected cartilages in our embryos. Based on this we do not expect a massive change in migratory crest cells' behavior. On the other side, identifying the genes/pathways that may be affected would be very difficult to visualize since they only affect a specific group of cells that will form those cartilages. Other cranial structures, such as tendons and/or muscles, may also be affected, causing changes in the shapes and lengths of the cartilage. We agree with the reviewer that identifying the affected genes or pathways would be very interesting, but we find it extremely difficult to sort the specific group of cells affected in the offspring in order to accurately analyze the expression levels of specific genes or to perform an RNAseq.

In view of established genome, transcriptome and proteome analysis techniques, it would be possible to carry out a more detailed analysis, for example of the transcriptome. This could shed light on the hypothesis that the reported alterations of specific SncRNAs cause the observed embryonic malformations.

As also suggested by reviewer 1, we did a more detailed analysis of the transcriptome, and now included rRNAs (Figure 3). However, we didn't observe in our study quantitative differences between conditions in both amount and size distribution. The new differential expression analysis now includes all 6 rRNA species (Figure 3C-D). As a consequence, Deseq2 results may vary slightly with respect to the old figures, which were all updated. This is because the normalization algorithm used by Deseq2 is affected by every row in the count matrix. However, it is important to mention that these changes are marginal and conclusions regarding changes in tsRNA expression due to methotrexate treatment remain the same.

Finally, to demonstrate unequivocally that sncRNA variations are the cause, in part, of the observed embryonic malformations. To this end we have now isolated from MTX- and Control-treated males small-RNA fractions (20-50 nt and 50-90 nt) and injected them independently and combined (20-90 nt) into fertilized wild-type eggs.

Our results show that injection of 20-50 nt, 50-90 nt or the combination of both (20-90 nt) at the two doses of MTX-treated sperm showed a significant reduction ($p < 0.0001$) on the ceratohyal lengths compared to control (Fig. 6A). On the other side, when we analyze the basihyal phenotypes we were unable to evidence any hook shape malformation, but instead we only evidenced embryos having curved trowel shape bended upward or downward (affected) (Fig. 6B). Importantly, injection of 20-50 nt fraction from both 10MTX and 50MTX, but not the 50-90 nt, significantly increase the number of larvae having affected basihyal shape (Fig. 6B). Similarly, injection of both 20-90 nt RNA-fractions have a similar effect than the 20-50 nt. All these together suggest that RNAs from exposed males have the ability to alter the development of specific cranial cartilages on the offspring.

Figure S1 D shows a delay of hatching of treated embryos. Is this significant? If yes, the authors should mention/discuss this “phenotype”.

Our analyze fails to evidence significant differences.

Reviewer #3:

First of all, we would like to express our gratitude to reviewer 3 for the positive criticisms. All the comments are very appropriated and constructive, which is highly appreciated. All the changes in this new version of the manuscript are highlighted in yellow.

Overall, this is a strong contribution. However, the authors should make more conservative conclusions. The changes in small non-coding RNAs may or may not lead to craniofacial defects. There was no evidence presented to show any causation between the expression changes and phenotype. I have specific comments below.

The observation of the reviewer is valid. Accordingly, we have now isolated from MTX- and Control- treated males small-RNA fractions (20-50 nt and 50-90 nt) and injected them independently and combined (20-90 nt) into fertilized wild-type eggs.

Our results show that injection of 20-50 nt, 50-90 nt or the combination of both (20-90 nt) at the two doses of MTX-treated sperm showed a significant reduction ($p < 0.0001$) on the ceratohyal lengths compared to control (Fig. 6A). On the other side, when we analyze the basihyal phenotypes we were unable to evidence any hook shape malformation, but instead we only evidenced embryos having curved trowel shape bended upward or downward (affected) (Fig. 6B). Importantly, injection of 20-50 nt fraction from both 10MTX and 50MTX, but not the 50-90 nt, significantly increase the number of larvae having affected basihyal shape (Fig. 6B). Similarly, injection of both 20-90 nt RNA-fractions have a similar effect than the 20-50 nt. All these together suggest that RNAs from exposed males have the ability to alter the development of specific cranial cartilages on the offspring.

1- The numbers of animal used to generate the statistics should be given in the results or figure legends.

Thanks for pointing this out. We have now included the number of analyzed embryos in the figures.

2. The light color of the bars in Figure 4 and 5 are very hard to see. The contrast between the bar color and the white page should be changed to improve visibility.

Colors have been changed for a better visualization of the figures.

3. Line 192. As the sentence reads, the authors give the impression that MTX binds to the enzyme activity. Removing the word “activity” would solve the issue.
Done.

4. Line 195. The sentence should read “and for pregnancy termination.”

Done.

5. Line 233. "Up to now" is stated like there is a difference after the findings in this paper. The tsRNA functions in this instance remain speculative. More conservative statements are needed here and elsewhere to avoid giving the impression that the data presented here connects the gene expression changes with phenotype changes. Causation was not established. There is only a correlation, which should be stated clearly.

Changes have been made following the reviewer suggestion.

Reviewer #1 (Remarks to the Author):

I'm satisfactory with the revision and support publication in Nature Communications

Reviewer #2 (Remarks to the Author):

the authors have addressed the comments and criticism and improved the manuscript accordingly.
The RNA injection adds a strong point to support their arguments

Reviewer #3 (Remarks to the Author):

The authors have addressed my concerns by the addition of new experiments showing effects of small RNAs.